# MI-NeRF: Learning a Single Face NeRF from Multiple Identities

## Abstract

In this work, we introduce a method that learns a single dynamic neural radiance field (NeRF) from monocular talking face videos of multiple identities. NeRFs have shown remarkable results in modeling the 4D dynamics and appearance of human faces. However, they require expensive per-identity optimization. To address this challenge, we introduce MI-NeRF (multi-identity NeRF), a single unified network that models complex non-rigid facial motion for multiple identities, using only monocular videos of arbitrary length. The core premise in our method is to learn the non-linear interactions between identity and non-identity specific information with a multiplicative module. By training MI-NeRF on multiple videos simultaneously, we significantly reduce the total training time, compared to standard single-identity NeRFs. Our model can be further personalized for a target identity. We demonstrate results for both facial expression transfer and talking face video synthesis.

## 1 Introduction

Capturing the 4D dynamics and appearance of non-rigid motion of humans has long been a challenge for both computer vision and graphics. This task has broad applications, ranging from AR/VR and video games to virtual communication and the movie industry, all of which require the creation of photorealistic videos of the human face. Earlier approaches relied on 3D morphable models (3DMM) (Garrido et al., 2015; 2014; Thies et al., 2016), while more recent methods have turned to generative adversarial networks (GAN) (Pumarola et al., 2020; Prajwal et al., 2020; Vougioukas et al., 2020). GANs learn representations of facial dynamics from large datasets, containing video clips from multiple identities. To disentangle latent factors of variation, such as identity and expression, some works have imposed multilinear structures (Wang et al., 2019; Georgopoulos et al., 2020), building on the ideas of TensorFaces (Vasilescu & Terzopoulos, 2002). Despite their success, most GANs operate in the 2D image space and they do not model the 3D face geometry.

Neural radiance fields (NeRF) have recently demonstrated photorealistic 3D modeling of both static (Mildenhall et al., 2020; Barron et al., 2021; 2022) and dynamic scenes (Pumarola et al., 2021; Li et al., 2021; 2022), making them a popular choice for modeling human faces from monocular videos (Gafni et al., 2020; Park et al., 2021a;b; Athar et al., 2022). Approaches that leverage a 3DMM prior and condition on expression parameters enable control of facial expressions, for applications such as expression transfer (Gafni et al., 2020) and lip syncing (Chatziagapi et al., 2023). Despite their high-quality results, NeRFs require expensive per-scene or per-identity optimization. Although some works have learned generic scene representations (Wang et al., 2021a; Chen et al., 2021; Trevithick & Yang, 2021; Yu et al., 2021; Kwon et al., 2021; Mu et al., 2023), they require static settings and/or multiple input views during training.

In this work, we propose MI-NeRF (multi-identity NeRF), a novel method that learns a single dynamic NeRF from monocular talking face videos of multiple identities. Using only a *single* network, it learns to model complex non-rigid human face motion, while disentangling identity and non-identity specific information. At the epicenter of our approach lies a multiplicative module that approximates the non-linear interactions between latent factors of variation, inspired by ideas that go back to TensorFaces (Vasilescu & Terzopoulos, 2002). This module learns a non-linear mapping of identity codes and facial expressions, based on the Hadamard product. To the best of our

knowledge, this is the first method that learns a single unified face NeRF from monocular videos of multiple identities.

Trained on multiple videos simultaneously, MI-NeRF significantly reduces the training time, compared to multiple standard single-identity NeRFs, by up to 90%, leading to a sublinear cost curve. Further personalization for a target identity requires only a few iterations and leads to a performance on par with the state-of-the-art for facial expression transfer and audio-driven talking face synthesis. Since MI-NeRF leverages information from multiple identities, it also works for very short video clips of only a few seconds length. We intend to release the source code upon acceptance of the paper.

In brief, our contributions are as follows:

- We introduce MI-NeRF, a novel method that learns a single dynamic NeRF from monocular talking face videos of multiple identities.
- We propose a multiplicative module to learn non-linear interactions between identity and non-identity specific information. We present two specific parameterizations for this module and provide their technical derivations.
- Our generic model can be further personalized for a target identity, achieving state-of-the-art performance for facial expression transfer and talking face video synthesis, requiring only a fraction of the total training time of standard single-identity NeRFs.

## 2 RELATED WORK

**Human Portrait Video Synthesis.** Earlier approaches for video synthesis and editing of human faces are based on 3DMMs (Garrido et al., 2015; 2014; Thies et al., 2016). A 3DMM (Blanz & Vetter, 1999) is a parametric model that can represent a face as a linear combination of principle axes for shape, texture, and expression, learned by principal component analysis (PCA). More recently, GAN-based networks have been proposed for video synthesis (Kim et al., 2018; Siarohin et al., 2019; Pumarola et al., 2020), as well as for audio-driven talking faces (Prajwal et al., 2020; Zhou et al., 2021; Vougioukas et al., 2020). GANs are trained on large datasets with video clips from multiple identities, learning diverse facial expressions and lip movements. However, they operate in a low resolution 2D image space and they cannot model the 3D face geometry. NeRFs (Mildenhall et al., 2020) have become very popular, since they can represent the 3D face geometry and appearance, and generate photorealistic videos (Gafni et al., 2020; Guo et al., 2021; Park et al., 2021a;b).

**Multilinear Factor Analysis of Faces.** Factors of variation, such as identity, expression, and illumination, affect the appearance of a human face in a portrait video. Disentangling those factors is challenging. Techniques like PCA can only find a single mode of variation (Turk & Pentland, 1991). TensorFaces (Vasilescu & Terzopoulos, 2002) is an early approach that approximates different modes of variation using a multilinear tensor decomposition. Inspired by this, several works have proposed to learn multiplicative interactions to disentangle latent factors of variation (Vlasic et al., 2006; Tang et al., 2013; Wang et al., 2017). Multilinear latent conditioning has also been proved beneficial for GANs and VAEs, in order to disentangle and edit face attributes (Sahasrabudhe et al., 2019; Wang et al., 2019; Georgopoulos et al., 2020; Chrysos et al., 2021). In this work, we propose a multiplicative module that conditions a NeRF and approximates the non-linear interactions between identity and non-identity specific information.

**Neural Radiance Fields.** Implicit neural representations for modeling 3D scenes have recently gained a lot of attention. In particular, NeRFs (Mildenhall et al., 2020; Barron et al., 2021; 2022; Lindell et al., 2022) have shown photorealistic novel view synthesis of complex scenes. They represent a static scene as a continuous 5D function, using a multilayer perceptron (MLP) that maps each 5D coordinate (3D spatial location and 2D viewing direction) to an RGB color and volume density. However, NeRFs require expensive per-scene or per-identity optimization. A few works have proposed to learn generic representations (Wang et al., 2021a; Chen et al., 2021; Trevithick & Yang, 2021; Yu et al., 2021), but these require static settings and multiple views as input. In contrast, in this work, we are interested in dynamic human faces, captured from monocular videos.

**Dynamic Neural Radiance Fields for Human Faces.** Several works have extended NeRFs to dynamic scenes (Pumarola et al., 2021; Li et al., 2021; 2022; Park et al., 2021a; Gafni et al., 2020;

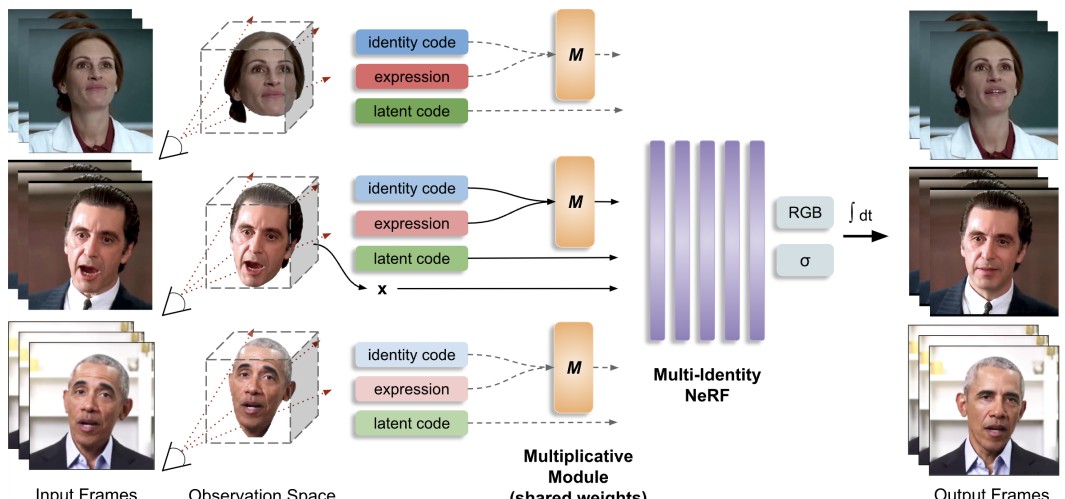

Figure 1: **Overview of MI-NeRF.** Given monocular talking face videos from multiple identities, MI-NeRF learns a *single* network to model their 4D geometry and appearance. A multiplicative module with shared weights across all identities learns non-linear interactions between identity codes and facial expressions. MI-NeRF can synthesize high-quality videos of any input identity.

Park et al., 2021b; Weng et al., 2022). They usually map the sampled 3D points from an observation space to a canonical space, in order to learn a time-invariant scene representation. Some additionally learn time-variant latent codes (Gafni et al., 2020; Li et al., 2022; Chatziagapi et al., 2023). Particularly challenging is to capture the 4D dynamics and appearance of the non-rigid deformations of the human face from monocular videos. Conditioning a NeRF on 3DMM expression parameters, related works enable explicit and meaningful control of the synthesized subject (Gafni et al., 2020; Guo et al., 2021; Athar et al., 2022; 2021; Chatziagapi et al., 2023). Although these approaches can produce HD quality results, they are identity-specific and usually require long videos for training. Only a limited number of prior works have aimed to train a generic NeRF for human faces (Raj et al., 2021; Hong et al., 2022; Zhuang et al., 2022). However, all of these models require multiple views for training. In contrast, we propose a simple architecture that is capable of learning multiple identities from monocular talking face videos of arbitrary length captured in the wild.

## 3 METHOD

We present MI-NeRF, a novel method that learns a single dynamic NeRF from monocular talking face videos of multiple identities. An overview of our approach is illustrated in Fig. 1. Given RGB videos of different subjects, we learn a *single* unified network that represents their 4D facial geometry and appearance. A multiplicative module approximates the non-linear interactions between learned identity codes and facial expressions, in order to disentangle identity and non-identity specific information. Its output, along with learned per-frame latent codes, condition a dynamic NeRF. With this simple architecture, MI-NeRF enables training a NeRF on a large number of human faces, reducing the total training time of standard single-identity NeRFs, and achieving high-quality video synthesis for any input subject.

### 3.1 CONDITIONAL INPUT

**Head Pose and Expression.** For each video frame of an identity, we fit a 3DMM and extract the corresponding head pose $P \in \mathbb{R}^{4 \times 4}$ and expression parameters $e \in \mathbb{R}^{79}$. We follow an optimization-based method, that minimizes an objective function with photo-consistency and landmark terms (Guo et al., 2018b). We use the learned axes from Guo et al. (2018b), based on the Basel Face Model (Paysan et al., 2009) for shape and texture and the FaceWarehouse (Cao et al., 2013) for expression. The extracted head pose $P = [R; t]$ is used to transform the sampled 3D points to the canonical space before shooting the rays, where $R \in \mathbb{R}^{3 \times 3}$ and $t \in \mathbb{R}^{3 \times 1}$ are the rotation and translation matrices correspondingly.

**Learned Identity and Latent Codes.** In addition to the expression vectors, the dynamic NeRF is conditioned on learned identity and latent codes. Both are randomly initialized embeddings that are learned during training. We use one identity code $\boldsymbol{i} \in \mathbb{R}^{79}$ per video [1], in order to capture *time-invariant* information. These codes appear to mainly capture the identity, and thus we call them identity codes. We also learn one latent code $\boldsymbol{l} \in \mathbb{R}^{32}$ per frame per video, in order to capture *time-varying* information. These latent codes memorize very small per-frame variations, such as appearance and illumination, that are independent of facial expressions, but necessary to reconstruct them in the output videos (Gafni et al., 2020; Chatziagapi et al., 2023).

## 3.2 Proposed Modules

We propose to learn *multiplicative interactions* between facial expressions and identity codes using the Hadamard product. Earlier works have used multilinear tensor decomposition to disentangle latent factors of variation of the human face (Vasilescu & Terzopoulos, 2002; Wang et al., 2019; Georgopoulos et al., 2020). Inspired by this, we learn a non-linear mapping that disentangles identity and non-identity specific information. This mapping is learned by a single multiplicative module $M$ with shared weights for all identities. We also introduce a variation of this module, $H$, that captures high-degree interactions. Please see the appendix for the detailed derivation.

### 3.2.1 Multiplicative Interaction Module

Given an expression vector $\boldsymbol{e} \in \mathbb{R}^d$ and an identity vector $\boldsymbol{i} \in \mathbb{R}^d$, our multiplicative module $M$ learns the following mapping:

$$M(\boldsymbol{e}, \boldsymbol{i}) = \boldsymbol{C}\left[(\boldsymbol{U}_1 \boldsymbol{e}) * (\boldsymbol{U}_2 \boldsymbol{i})\right] + \boldsymbol{W}_2 \boldsymbol{e} + \boldsymbol{W}_3 \boldsymbol{i} , \tag{1}$$

where $*$ denotes the Hadamard (element-wise) product that correlates $\boldsymbol{e}$ and $\boldsymbol{i}$, $\boldsymbol{U}_1 \in \mathbb{R}^{k \times d}$, $\boldsymbol{U}_2 \in \mathbb{R}^{k \times d}$, $\boldsymbol{C} \in \mathbb{R}^{d \times k}$, $\boldsymbol{W}_2 \in \mathbb{R}^{d \times d}$, $\boldsymbol{W}_3 \in \mathbb{R}^{d \times d}$ are learnable parameters, and $d = 79$ for our case. We chose $k < d$ to get low rank matrices, with fewer parameters. We experimentally found that this mapping $M$ with $k = 8$ leads to the best disentanglement between identity and expression with the least number of parameters (see ablation study in Sec. 4.2 for more details). Prop. 1 verifies the multiplicative interactions learned; its proof exists in Appendix A.1.

**Proposition 1.** *The function $M$ of Eq.* (1) *captures multiplicative interactions.*

### 3.2.2 High-degree Interaction Module

We can extend the multiplicative interaction module further, in order to capture high-degree interactions. Instead of directly multiplying the embeddings of the expression and the identity vector, we can find a common embedding space, add their features together and then perform multiplications. The formula of this module for the expression $\boldsymbol{e}$ and identity $\boldsymbol{i}$ vectors is the following:

$$H(\boldsymbol{e}, \boldsymbol{i}) = \boldsymbol{C}\boldsymbol{x}_N , \text{ where } \boldsymbol{x}_n = \boldsymbol{x}_{n-1} + \left(\boldsymbol{U}_{(n,1)}\boldsymbol{e} + \boldsymbol{U}_{(n,2)}\boldsymbol{i}\right) * \boldsymbol{x}_{n-1} , \tag{2}$$

for $n = 2, \ldots, N$, with $\boldsymbol{x}_1 = \boldsymbol{U}_{(1,1)}\boldsymbol{e} + \boldsymbol{U}_{(1,2)}\boldsymbol{i}$. The parameters $\boldsymbol{U}_{(n,1)} \in \mathbb{R}^{k \times d}$, $\boldsymbol{U}_{(n,2)} \in \mathbb{R}^{k \times d}$, $\boldsymbol{C} \in \mathbb{R}^{o \times k}$ are learnable for $n = 1, \ldots, N$. In practice, we chose $d = k = o = 79$, and $N = 2$ for our experiments, but capturing high-degree interactions might be beneficial in other cases. Prop. 2 and Prop. 3 in Appendix A.2 demonstrate the interactions learned in this case.

## 3.3 Dynamic NeRF

To model the dynamics of human faces, we learn a single dynamic NeRF for *all* input identities. For each video frame of a subject, we first segment the head using an automatic parsing method (Lee et al., 2020), similarly with Gafni et al. (2020); Guo et al. (2021); Chatziagapi et al. (2023). Then, we learn an implicit representation $F_\Theta$ of the identities using an MLP. Given an identity $\boldsymbol{i}$ at a specific video frame, shown from a particular viewpoint and with a particular facial expression, we first march camera rays through the scene and sample 3D points on these rays. For a 3D point location

---

[1]In our preliminary experiments, we found that defining the identity vector with the same dimension as the expression vector was sufficient.

$\boldsymbol{x}$, a viewing direction $\boldsymbol{v}$, the estimated expression vector $\boldsymbol{e}$, and a learned latent vector $\boldsymbol{l}$, $F_\Theta$ predicts the RGB color $\boldsymbol{c}$ and density $\sigma$ of the point:

$$F_\Theta : (M(\boldsymbol{e}, \boldsymbol{i}), \boldsymbol{l}, \boldsymbol{x}, \boldsymbol{v}) \longrightarrow (\boldsymbol{c}, \sigma) . \tag{3}$$

Given the predicted color $\boldsymbol{c}$ and density $\sigma$ for every point on each ray, we can produce the final video frame applying volumetric rendering (Mildenhall et al., 2020). For each camera ray $\boldsymbol{r}(t) = \boldsymbol{o} + t\boldsymbol{v}$ with camera center $\boldsymbol{o}$ and viewing direction $\boldsymbol{v}$, the color $C$ of the corresponding pixel can be computed by accumulating the predicted colors and densities of the sampled points along the ray:

$$C(\boldsymbol{r}; \Theta) = \int_{t_n}^{t_f} \sigma(\boldsymbol{r}(t)) \boldsymbol{c}(\boldsymbol{r}(t), \boldsymbol{v}) T(t) dt , \tag{4}$$

where $t_n$ and $t_f$ are the near and far bounds correspondingly, and $T(t) = \exp\left(-\int_{t_n}^{t} \sigma(\boldsymbol{r}(s)) ds\right)$ is the accumulated transmittance along the ray from $t_n$ to $t$.

Similarly to NeRF (Mildenhall et al., 2020), we follow a hierarchical sampling strategy, optimizing a coarse and a fine model. During training, we minimize the following objective function:

$$\mathcal{L} = \mathcal{L}_c + \lambda_l \mathcal{L}_l + \lambda_i \mathcal{L}_i , \text{ where } \mathcal{L}_c = \sum_{\boldsymbol{r}} \left\| \hat{C}(\boldsymbol{r}; \Theta) - C(\boldsymbol{r}) \right\|_2^2 \tag{5}$$

is the photo-consistency loss that measures the pixel-level difference between the ground truth color $C(\boldsymbol{r})$ and the predicted color $\hat{C}(\boldsymbol{r}; \Theta)$ for all the rays $\boldsymbol{r}$, $\mathcal{L}_l = \|\boldsymbol{l}\|_2$ and $\mathcal{L}_i = \|\boldsymbol{i}\|_2$ regularize the latent and identity vectors correspondingly, $\lambda_l = 0.01$ and $\lambda_i = 10^{-4}$.

**Implementation Details.** We use an MLP of 8 linear layers with a hidden size of 256 and ReLU activations, with branches for $\boldsymbol{c}$ and $\sigma$, positional encodings for $\boldsymbol{x}$ and $\boldsymbol{v}$ of 10 and 4 frequencies respectively, and Adam optimizer (Kingma & Ba, 2014) with a learning rate of $5 \times 10^{-4}$ that decays exponentially to $5 \times 10^{-5}$ (see appendix D for more details).

## 3.4 PERSONALIZATION

Trained on multiple identities simultaneously, MI-NeRF learns a large variety of facial expressions from diverse human faces and can synthesize videos of any training identity. To enhance the visual quality for a particular seen subject, i.e. to better capture their facial details, such as wrinkles, we can further improve the output appearance using a short video of this subject. We call this procedure "personalization". More specifically, we fine-tune the network with a small learning rate ($10^{-5}$) for only a few iterations, keeping the weights of the multiplicative module frozen. This idea can also be used to adapt MI-NeRF to an unseen identity (that is not part of the initial training set). Given only a few frames of an unseen subject, we can fine-tune our network to learn their identity and latent codes. Then, we can synthesize high-quality videos of them, given novel expressions as input.

## 4 EXPERIMENTS

### 4.1 DATASET AND EVALUATION

**Dataset.** To evaluate our proposed method, we collected 140 talking face videos of different identities from publicly available datasets (Guo et al., 2021; Lu et al., 2021; Chatziagapi et al., 2023; Hazirbas et al., 2021; Ginosar et al., 2019; Ahuja et al., 2020; Duarte et al., 2021; Zhang et al., 2021; Wang et al., 2021b). We included a variety of standard front-facing videos (e.g. political speeches), as well as more challenging videos, with large variations in head pose, lighting, and expressiveness (e.g. movies and news satire television programs). The videos are in HD quality (720p) and of around 20 seconds to 5 minutes duration. For each video, we run the 3DMM fitting procedure, as described in Sec. 3.1. We use 100 videos for our training set and we keep the rest as novel identities.

**Evaluation Metrics.** We measure the visual quality of the generated videos, using peak signal-to-noise ratio (PSNR), structural similarity index (SSIM) (Wang et al., 2004), and learned perceptual image patch similarity (LPIPS) (Zhang et al., 2018), and we verify the identity of the target subject, using the average content distance (ACD) (Vougioukas et al., 2019; Tulyakov et al., 2018). Additionally, we use the LSE-D (Lip Sync Error - Distance) and LSE-C (Lip Sync Error - Confidence) metrics (Prajwal et al., 2020; Chung & Zisserman, 2016), to assess the lip synchronization, i.e. if the generated expressions are meaningful given the corresponding speech signal.

Table 1: **Ablation Study.** Quantitative results for different variants of our model. The proposed multiplicative module $M$ leads to the best disentanglement (lower ACD) and visual quality (higher PSNR) with the least possible parameters.

| Method | PSNR ↑ | ACD ↓ | LSE-D ↓ | LSE-C ↑ |
|---|---|---|---|---|
| Baseline NeRF (without $M$) | 28.65 | 0.229 | 9.08 | 4.06 |
| (A1) $M(e, i) = W_2 e + W_3 i$ | 29.08 | 0.200 | 8.80 | 4.19 |
| (A2) $M(e, i) = (e * i) + W_2 e + W_3 i$ | 28.84 | 0.207 | 9.07 | 3.80 |
| (A3) $M(e, i) = W_1 (e * i) + W_1 e + W_1 i$ | 28.50 | 0.227 | 9.19 | 3.61 |
| (A4) $M(e, i) = W_1 (e * i)$ | 29.12 | 0.228 | 10.49 | 2.36 |
| (A5) $M(e, i) = (W_2 e) * (W_3 i) + W_2 e + W_3 i$ | 28.95 | 0.204 | 9.25 | 3.58 |
| (A6) $M(e, i) = W_1 (e * i) + W_2 e + W_3 i$ | 28.53 | 0.221 | 9.27 | 3.29 |
| (A7) $M(e, i) = \mathcal{W} \times_2 e \times_3 i$ | 28.41 | 0.274 | 9.82 | 3.10 |
| MI-NeRF with $M$ without latent codes $l$ | 29.01 | 0.165 | 8.83 | 4.15 |
| MI-NeRF with $M$ with $k = 32$ | 29.02 | 0.180 | 8.79 | 4.21 |
| MI-NeRF with $H$ with $N = 4$ | 28.00 | 0.289 | 10.76 | 2.19 |
| MI-NeRF with $M$ (Ours) | **29.73** | **0.158** | **8.62** | **4.24** |
| MI-NeRF with $H$ (Ours) | 28.95 | 0.180 | 8.82 | **4.24** |

## 4.2 ABLATION STUDY

We conduct an ablation study on the multiplicative module and the input conditions of MI-NeRF. Given 10 videos from different identities, we investigate variants of our model (see Table 1). After training, we synthesize a video of each identity given input expressions from another one. In this way, we evaluate if the model learns to disentangle identity and non-identity specific information.

Firstly, we evaluate the possibility of simply concatenating all the input vectors, as usually done in standard identity-specific NeRFs (Gafni et al., 2020; Guo et al., 2021; Chatziagapi et al., 2023), i.e. omitting the multiplicative module. We call this "Baseline NeRF". As shown in Table 1 and Fig. 2, Baseline NeRF cannot learn to disentangle between different identities. It frequently synthesizes a different identity than the target one, or a mixture of identities, based on the identity-expression pairs seen during training.

Variants of the proposed multiplicative module $M$ are examined in rows (A1) to (A7) to determine the simplest and most effective option. We started by just learning a simple mapping of the $e$ and $i$ vectors, using $W_2$ and $W_3$ matrices (A1). We then explored variants that include a Hadamard product $(e * i)$ to model their non-linear interaction. The last variant (A7) is inspired by Wang et al. (2019) and uses $\mathcal{W} \in \mathbb{R}^{d \times d \times d}$, requiring significantly more parameters than our proposed $M$ ($d^3$ vs $d^2$). The variant (A6) can be derived from (A7), using similar arguments to Appendix A.1. All these variants (A1) - (A7) lead to different disentanglement results, with many of them being quite poor.

In the following rows, we show the small decrease in visual quality if we omit the latent codes, that learn very small per-frame variations in appearance, as noted in Sec. 3.1, or use different hyperparameters: $k = 32$ for $M$ and $N = 4$ for $H$. We conclude that our proposed multiplicative module $M$ leads to the best disentanglement and highest visual quality with the least possible parameters. Our proposed $H$ leads to the second best disentanglement between identity and expression (low ACD metric) and produces videos of comparable visual quality and lip synchronization (LSE metrics).

## 4.3 FACIAL EXPRESSION TRANSFER

In this section, we demonstrate the effectiveness of MI-NeRF for facial expression transfer. Given an identity from the training set, MI-NeRF enables explicit control of their expressions, and thus can synthesize high-quality videos of them given novel expressions as input.

**Results.** Fig. 2 demonstrates the result of MI-NeRF with $M$ (personalized) for challenging novel expressions, not seen for the target identity. Fig. 3 (right) shows the corresponding PSNR, computed on frames with challenging unseen expressions for a particular subject. MI-NeRF significantly outperforms the standard single-identity NeRFace (Gafni et al., 2020) in such cases. Increasing the number of training identities improves the robustness of our generic model. Further personalization enhances the final visual quality.

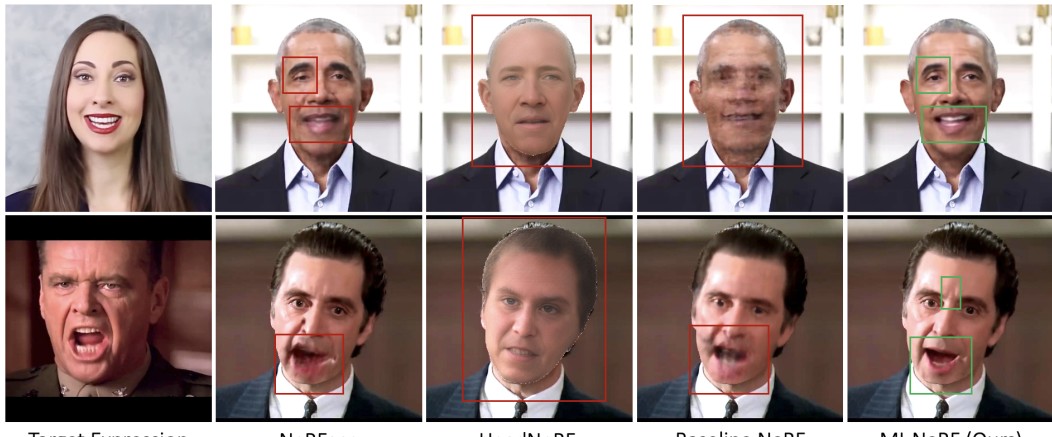

Figure 2: **Transferring Novel Expressions.** Qualitative comparison of MI-NeRF with state-of-the-art approaches for transferring unseen expressions to a target identity. NeRFace (Gafni et al., 2020) is a single-identity NeRF, HeadNeRF (Hong et al., 2022) is a NeRF-based parametric head model, Baseline NeRF concatenates all input conditions, leading to poor disentanglement. Our method demonstrates robustness in synthesizing novel expressions for any input identity.

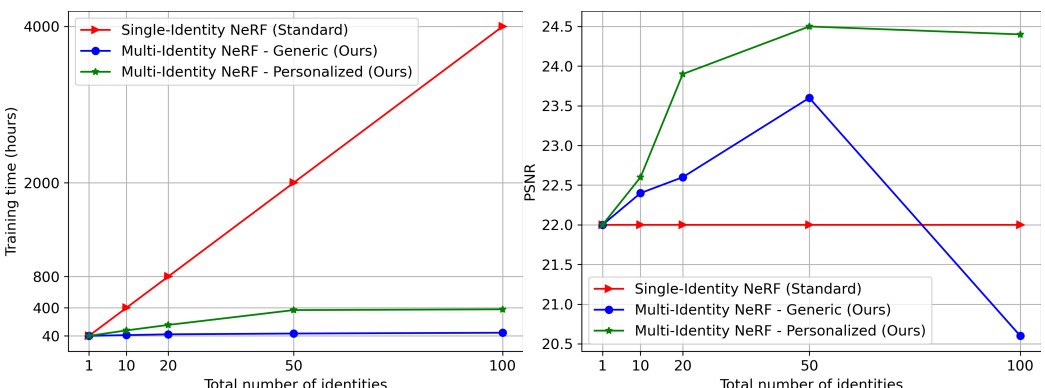

Figure 3: **Left:** Training time vs total number of identities. Standard single-identity NeRFs, like NeRFace (Gafni et al., 2020), AD-NeRF (Guo et al., 2021), and LipNeRF (Chatziagapi et al., 2023) require approximately 40 hours training per identity. On the contrary, our MI-NeRF (generic) can be trained on 100 identities in 80 hours, a $50\times$ improvement over the previous methods. Further personalization takes another 5-8 hours per identity approximately. **Right:** Corresponding visual quality of generated videos with challenging novel expressions, measured by PSNR (higher the better). Increasing the number of identities improves the robustness of our model to unseen expressions.

Table 2 (left) shows the quantitative results of facial expression transfer, computed on 10 synthesized videos (of 10 distinct identities, given a sequence of expressions from a different identity). We compare with methods that similarly allow meaningful control of expression parameters. NeRFace (Gafni et al., 2020) is a standard identity-specific NeRF, without any multiplicative module. Its immediate extension is the Baseline NeRF that is trained on multiple identities, concatenating an additional identity vector as input. HeadNeRF (Hong et al., 2022) is a novel NeRF-based parametric head model, trained on a large amount of high-quality images from multiple identities.

We notice that NeRFace can produce good visual quality, but cannot generalize to unseen expressions. Trained only on a video of a single identity, the model has only seen a limited variety of facial expressions, and thus can lead to artifacts in case of novel input expressions (see second row in Fig. 2). Baseline NeRF, as also mentioned in the previous section, leads to poor disentanglement and frequently generates a mixture of identities, depending on the input expression. HeadNeRF distorts the target identity and inaccurately produces the target facial expression. Our method demon-

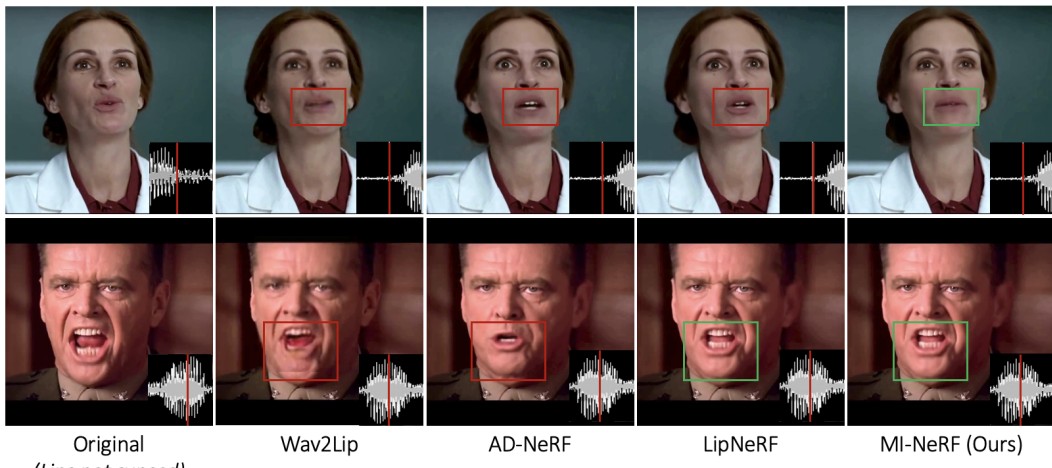

| Original | Wav2Lip | AD-NeRF | LipNeRF | MI-NeRF (Ours) |

*(Lips not synced)*

Figure 4: **Lip Synced Video Synthesis.** Qualitative comparison of our method with state-of-the-art approaches, GAN-based Wav2Lip (Prajwal et al., 2020), AD-NeRF (Guo et al., 2021) and Lip-NeRF (Chatziagapi et al., 2023). The original video is in English (1st column). The generated videos (columns 2-5) are lip synced to dubbed audio in Spanish.

Table 2: **Left:** Facial Expression Transfer. Quantitative comparison of our method with state-of-the-art methods for transferring facial expressions. **Right:** Lip Synced Video Synthesis. Quantitative comparison of our method with state-of-the-art approaches on generated videos lip-synced to dubbed audio in different languages.

| Method | PSNR ↑ | SSIM ↑ | LPIPS ↓ | ACD ↓ | Method | LSE-D ↓ | LSE-C ↑ | PSNR ↑ | SSIM ↑ |
|---|---|---|---|---|---|---|---|---|---|
| NeRFace | 28.11 | **0.88** | **0.13** | **0.11** | AD-NeRF | 11.40 | 1.28 | 27.38 | 0.84 |
| HeadNeRF | 19.89 | 0.74 | 0.32 | 0.85 | LipNeRF | 9.92 | 2.71 | 30.10 | 0.89 |
| Baseline NeRF | 26.21 | 0.85 | 0.22 | 0.32 | GeneFace | 11.80 | 2.22 | 25.56 | 0.85 |
| MI-NeRF (Ours) | **28.28** | **0.88** | **0.13** | **0.11** | MI-NeRF (Ours) | **9.46** | **2.98** | **30.21** | **0.90** |

strates robustness in synthesizing novel expressions for any identity, as it leverages information from multiple subjects during training. We encourage the readers to watch the supplementary video.

**Training Time.** MI-NeRF significantly outperforms the standard single-identity NeRF-based methods, like NeRFace (Gafni et al., 2020), in terms of the training time (see Fig. 3 left). Training on 100 identities needs only about 80 hours of training, compared to 40 hours per identity needed by standard NeRFs, leading to a 90% decrease approximately. Further personalization for a target identity adds only another 5-8 hours of training on average, enhancing the visual quality (see Fig. 3 right).

## 4.4 LIP SYNCED VIDEO SYNTHESIS

MI-NeRF can be also used for audio-driven talking face video synthesis, also called lip-syncing. Prior work, LipNeRF (Chatziagapi et al., 2023), has shown that conditioning a NeRF on the 3DMM expression space, compared to audio features, leads to more accurate and photorealistic lip synced videos. Our method naturally extends LipNeRF to multiple identities, using their proposed audio-to-expression mapping.

For this task, we used the dataset proposed by LipNeRF (Chatziagapi et al., 2023) [2]. It includes 10 videos from popular movies in English, and corresponding dubbed audio in 2 or 3 different languages for each video, including French, Spanish, German and Italian.

**Results.** Table 2 (right) shows the corresponding quantitative evaluation of the synthesized videos, lip synced to dubbed audio in different languages. Leveraging information from multiple identities,

---

[2]We would like to thank the authors of LipNeRF for providing the cinematic data from YouTube.

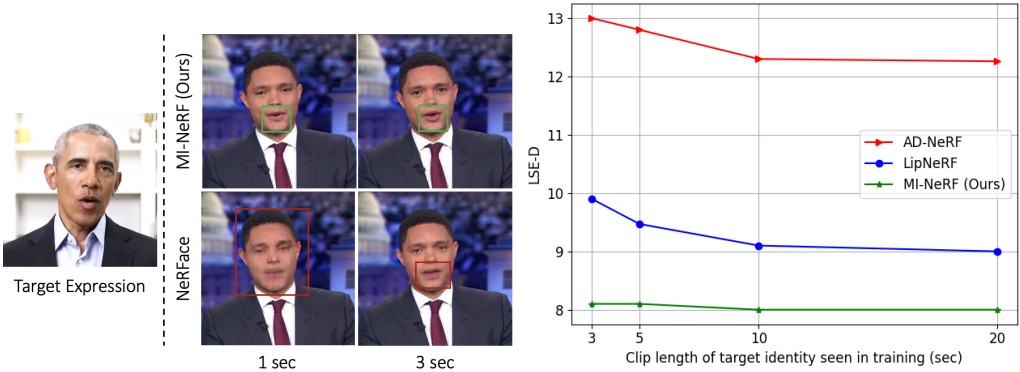

Figure 5: **Short-Video Personalization. Left:** Qualitative comparison of MI-NeRF with the single-identity NeRFace (Gafni et al., 2020), using a video of the target identity of only 1 or 3 seconds duration for adaptation. **Right:** LSE-D (lower the better) vs clip length of the target identity seen in training, computed on generated videos lip synced to dubbed audio.

MI-NeRF slightly improves LSE metrics and achieves similar visual quality with LipNeRF, at much lower training time (see Fig. 3). AD-NeRF (Guo et al., 2021) and GeneFace (Ye et al., 2023) lack in lip synchronization for this task.

Fig. 4 compares the mouth position for two examples, lip synced to Spanish (original audio in English). The GAN-based method, Wav2Lip (Prajwal et al., 2020), frequently produces artifacts and blurry results. Since it operates in the 2D image space, it cannot handle large 3D movements. AD-NeRF overfits to the training audio and cannot generalize well to different speech inputs. LipNeRF performs well, but sometimes cannot produce unseen expressions (e.g. does not close the mouth in the first row), since it is only trained on a single video of limited duration. Both AD-NeRF and LipNeRF are standard single-identity NeRFs, requiring expensive identity-specific optimization.

## 4.5 SHORT-VIDEO PERSONALIZATION

MI-NeRF can also be adapted to an unseen identity, i.e. not seen as a part of the initial training set (see Sec. 3.4). Fig. 5 (left) demonstrates the results, when only a very short video of the new identity is available (1 or 3 seconds length). Please note that in this case we use a small number of *consecutive* frames, and thus a very small part of the expression space is covered. In contrast to NeRFace, MI-NeRF produces satisfactory lip shape and expression, as the multiplicative module has learned information from multiple identities during training. Correspondingly, Fig. 5 (right) shows the LSE-D metric w.r.t different clip lengths of the target identity for the task of lip-syncing. Since AD-NeRF and LipNeRF are trained on a single identity, they learn audio-lip representations from only 3 or 20 seconds. On the other hand, MI-NeRF leverages information from multiple identities and leads to a more accurate lip synchronization.

## 5 CONCLUSION

In this work, we introduce MI-NeRF that learns a single dynamic NeRF from monocular talking face videos of multiple identities. We propose a multiplicative module that captures the non-linear interactions of identity and non-identity specific information. Trained on multiple identities, MI-NeRF significantly reduces the training time over standard single-identity NeRFs. Our model can be further personalized for a target identity, given only a short video of a few seconds length, achieving state-of-the-art performance for facial expression transfer and talking face video synthesis. In the future, we envision extending our approach to thousands of identities, learning collectively from very short in-the-wild video clips.

REPRODUCIBILITY STATEMENT

Our plan is to make the source code of our model publicly available once our work is accepted. We provide comprehensive documentation of the hyperparameters employed and offer detailed explanations of all the techniques used, supported by thorough ablation studies.

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

CONTENTS OF THE APPENDIX

The appendix is organized as follows:

- Technical Derivations of the Modules in Sec. A.
- Additional Ablation Study in Sec. B.
- Additional Results in Sec. C.
- Implementation Details in Sec. D.
- Ethical Considerations in Sec. E.
- Dataset Details in Sec. F.

We strongly encourage the readers to watch our supplementary video.

## A   TECHNICAL ANALYSIS OF THE MODELS

In this section, we complete the proofs for the model derivation. Firstly, let us establish a more detailed notation.

**Notation**: Tensors are symbolized by calligraphic letters, e.g., $\boldsymbol{\mathcal{X}}$. The *mode-$m$ vector product* of $\boldsymbol{\mathcal{X}}$ with a vector $\boldsymbol{u} \in \mathbb{R}^{I_m}$ is denoted by $\boldsymbol{\mathcal{X}} \times_m \boldsymbol{u}$. A core tool in our analysis is the CP decomposition Kolda & Bader (2009). By considering the mode-1 unfolding of an $M^{\text{th}}$-order tensor $\boldsymbol{\mathcal{X}}$, the CP decomposition can be written in matrix form as in Kolda & Bader (2009):

$$\boldsymbol{X}_{[1]} \doteq \boldsymbol{U}_{(1)} \bigg( \overset{2}{\underset{m=M}{\bigodot}} \boldsymbol{U}_{(m)} \bigg)^T , \tag{6}$$

where $\{\boldsymbol{U}_{(m)}\}_{m=1}^M$ are the factor matrices and $\odot$ denotes the Khatri-Rao product.

### A.1   PROOF OF PROP. 1

To prove the Proposition, we will construct the special form from the general multiplicative interaction.

The full multiplicative interaction between an expression vector $\boldsymbol{e} \in \mathbb{R}^d$ and an identity vector $\boldsymbol{i} \in \mathbb{R}^d$ is described by the following formula:

$$M^\dagger(\boldsymbol{e}, \boldsymbol{i}) = \boldsymbol{\mathcal{W}} \times_2 \boldsymbol{e} \times_3 \boldsymbol{i} , \tag{7}$$

where $\boldsymbol{\mathcal{W}} \in \mathbb{R}^{o \times d \times d}$ is a learnable tensor. We can also add the linear interactions in this model, which augment the equation as follows:

$$M^\dagger(\boldsymbol{e}, \boldsymbol{i}) = \boldsymbol{\mathcal{W}} \times_2 \boldsymbol{e} \times_3 \boldsymbol{i} + \boldsymbol{W}_2 \boldsymbol{e} + \boldsymbol{W}_3 \boldsymbol{i} , \tag{8}$$

where $\boldsymbol{W}_2, \boldsymbol{W}_3 \in \mathbb{R}^{o \times d}$ are learnable parameters. We can use the CP decomposition (Kolda & Bader, 2009) to induce a low-rank decomposition on the third-order tensor $\boldsymbol{\mathcal{W}}$. This results in the following expression:

$$M^\dagger(\boldsymbol{e}, \boldsymbol{i}) = \boldsymbol{C} \left( \boldsymbol{A} \odot \boldsymbol{B} \right)^T \left( \boldsymbol{e} \odot \boldsymbol{i} \right) + \boldsymbol{W}_2 \boldsymbol{e} + \boldsymbol{W}_3 \boldsymbol{i} , \tag{9}$$

where $\boldsymbol{C} \in \mathbb{R}^{o \times k}$ and $\boldsymbol{A}, \boldsymbol{B} \in \mathbb{R}^{d \times k}$ are learnable parameters. The symbol $k$ is the rank of the decomposition, while $\odot$ denotes the Khatri-Rao product.

We can then apply the mixed product property, and we obtain the following expression:

$$M^\dagger(\boldsymbol{e}, \boldsymbol{i}) = \boldsymbol{C} \left[ \left( \boldsymbol{A}^T \boldsymbol{e} \right) * \left( \boldsymbol{B}^T \boldsymbol{i} \right) \right] + \boldsymbol{W}_2 \boldsymbol{e} + \boldsymbol{W}_3 \boldsymbol{i} , \tag{10}$$

where $*$ is the Hadamard product. If we set $o = k = d$ and set $A^T = \boldsymbol{U}_1, B = \boldsymbol{U}_2$, then we end up with the model of Eq. (1) of the main paper, which concludes the proof.

## A.2 Intuition on Eq. (2)

The model of Eq. (2) can also capture multiplicative interactions in case of $N = 2$ or high-degree interactions in case $N > 2$. We provide some constructive proof to show the case of $N = 2$ below, since the number of terms increases fast already for this case. Subsequently, we focus on the case of high-degree interactions.

**Proposition 2.** *For $N = 2$, the model of Eq.* (2) *captures multiplicative interactions between the expression $e$ and the identity $i$ vector.*

*Proof.* For $N = 2$, Eq. (2) becomes:

$$H(e,i) = C\left\{\left(U_{(2,1)}e + U_{(2,2)}i\right) * \left(U_{(1,1)}e + U_{(1,2)}i\right)\right\} + C\left(U_{(1,1)}e + U_{(1,2)}i\right) =$$
$$C\left\{\left(U_{(2,1)}e\right) * \left(U_{(1,1)}e\right)\right\} + C\left\{\left(U_{(2,1)}e\right) * \left(U_{(1,2)}i\right)\right\} + C\left\{\left(U_{(2,2)}i\right) * \left(U_{(1,1)}e\right)\right\} + \quad (11)$$
$$C\left\{\left(U_{(2,2)}i\right) * \left(U_{(1,2)}i\right)\right\} + CU_{(1,1)}e + CU_{(1,2)}i \,.$$

Each of the first terms of the last equation arises from a CP decomposition with specific factor matrices (the inverse process from Eq. (8) to Eq. (9) can be followed). Therefore, Eq. (11) captures multiplicative interactions between the two variables, including second-degree interactions among the same variable. $\qquad\square$

**Proposition 3.** *For $N > 2$, the model of Eq.* (2) *captures high-degree interactions between the expression $e$ and the identity $i$ vector.*

*Proof.* The number of terms increase rapidly in Eq. (2). Without loss of generality, (a) we will only use the multiplicative term (and ignore the additive term of $+x_{n-1}$, since this is not contributing to the higher-degree terms) and (b) we will showcase this for $N = 3$. For $N = 3$, we obtain the following formula:

$$H^\dagger(e,i) \approx C\left\{\left(U_{(2,1)}e + U_{(2,2)}i\right) * \left(U_{(1,1)}e + U_{(1,2)}i\right) * \left(U_{(3,1)}e + U_{(3,2)}i\right)\right\} =$$
$$C\left\{\left(U_{(2,1)}e\right) * \left(U_{(1,1)}e\right) * \left(U_{(3,1)}e\right)\right\} + C\left\{\left(U_{(2,1)}e\right) * \left(U_{(1,2)}i\right) * \left(U_{(3,1)}e\right)\right\} +$$
$$C\left\{\left(U_{(2,2)}i\right) * \left(U_{(1,1)}e\right) * \left(U_{(3,1)}e\right)\right\} + C\left\{\left(U_{(2,1)}e\right) * \left(U_{(1,1)}e\right) * \left(U_{(3,2)}i\right)\right\} + \quad (12)$$
$$C\left\{\left(U_{(2,1)}e\right) * \left(U_{(1,2)}i\right) * \left(U_{(3,2)}i\right)\right\} + C\left\{\left(U_{(2,2)}i\right) * \left(U_{(1,1)}e\right) * \left(U_{(3,2)}i\right)\right\} +$$
$$C\left\{\left(U_{(2,2)}i\right) * \left(U_{(1,2)}i\right) * \left(U_{(3,1)}e\right)\right\} + C\left\{\left(U_{(2,2)}i\right) * \left(U_{(1,2)}i\right) * \left(U_{(3,2)}i\right)\right\} \,.$$

Following similar arguments as the proofs above and the unfolding of the CP decomposition, we can exhibit that all of those terms arise from CP decompositions with specific factor matrices, while each one captures a triplet of the form $(\tau_1, \tau_2, \tau_3)$ with $\tau \in \{e, i\}$. $\qquad\square$

## B Additional Ablation Study

In this section, we evaluate other variants of the conditional input of the NeRF (see Table 3 and Sec. 4.2).

**(a) Higher Output Dimension.** A first variant is to increase the output dimension of our multiplicative module $M$, with $C \in \mathbb{R}^{o \times k}$ and $o > d$. We tried $o = 256$ that might give a more informative input to our NeRF. However, we found that this leads to a similar performance, while increasing the learnable parameters.

**(b) Learnable Concatenation.** As a second variant, we tried to concatenate the information captured from the expression and identity codes: $M(e, i) = [W_2 e; W_3 i]$. Since this variant does not learn any multiplicative interactions between $e$ and $i$, it leads to a decrease in performance.

**(c) Latent Codes in $M$.** Another possible variant is to include the per-frame latent codes $l$ in our multiplicative module. In this way, we would learn multiplicative interactions between all three attributes $e, i, l \in \mathbb{R}^d$ as follows:

$$M(e,i,l) = C[(U_1 e) * (U_2 i) * (U_3 l)$$
$$+ (U_1 e) * (U_2 i) + (U_1 e) * (U_3 l) + (U_2 i) * (U_3 l) \quad (13)$$
$$+ (U_1 e) + (U_2 i) + (U_3 l)] \,,$$

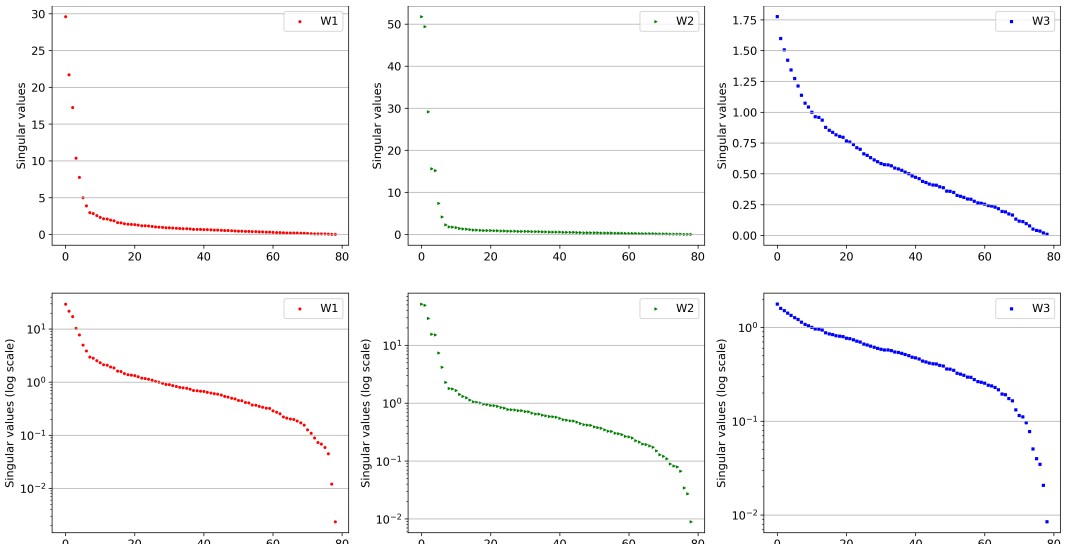

Figure 6: The singular values of our learned $\boldsymbol{W}_1, \boldsymbol{W}_2, \boldsymbol{W}_3 \in \mathbb{R}^{d \times d}$ of the (A6) variant of our multiplicative module (see Sec. 4.2), with $d = 79$, plotted in linear scale (first row) and logarithmic scale (second row). We found that $\boldsymbol{W}_1, \boldsymbol{W}_2, \boldsymbol{W}_3$ have full rank.

| Method | PSNR ↑ | ACD ↓ |
|---|---|---|
| (a) Output Dimension $o = 256$ | 29.76 | 0.16 |
| (b) Learnable Concatenation | 28.99 | 0.20 |
| (c) Latent Codes in $M$ | 29.01 | 0.17 |
| $M$ (Ours) | 29.73 | 0.16 |

Table 3: **Ablation Study.** Quantitative results for different variants of our conditional input: (a) $M$ with higher output dimension $o = 256$, (b) learnable concatenation without multiplicative interactions, and (c) latent codes included in the multiplicative module. The proposed multiplicative module $M$ leads to the best disentanglement with the least possible parameters.

where $\boldsymbol{U}_1, \boldsymbol{U}_2, \boldsymbol{U}_3 \in \mathbb{R}^{k \times d}$ and $\boldsymbol{C} \in \mathbb{R}^{o \times k}$. We set $o = k = d$. In this case, we capture third-order multiplicative interactions as well. The implicit representation $F_\Theta$ of the dynamic NeRF (see Eq. 3 of the main paper) would be:

$$F_\Theta : (M(\boldsymbol{e}, \boldsymbol{i}, \boldsymbol{l}), \boldsymbol{x}, \boldsymbol{v}) \longrightarrow (\boldsymbol{c}, \sigma) \tag{14}$$

We found that this variant would decrease the performance, making more difficult for the model to disentangle between identity and non-identity specific information. We believe that this happens because the latent codes $\boldsymbol{l}$ capture time-varying information. They memorize small per-frame variations in appearance for each video. These variations are reconstructed in the synthesized videos, in order to enhance the output visual quality. However, there are no meaningful interactions to learn between this time-varying information and the time-invariant identity codes or the facial expressions.

**Full Rank Matrices.** For the (A6) variant of our multiplicative module (see Sec. 4.2), we found that the learned $\boldsymbol{W}_1, \boldsymbol{W}_2, \boldsymbol{W}_3 \in \mathbb{R}^{d \times d}$ with $d = 79$ have full rank. Fig. 6 plots the 79 singular values of $\boldsymbol{W}_1, \boldsymbol{W}_2, \boldsymbol{W}_3$ from our model trained on 10 identities (applying SVD from numpy.linalg [3]). In the second row, we plot them in logarithmic scale. All of them have 79 singular values greater than $10^{-3}$, and until the 78th are far from zero, leading to matrices of full rank. This indicates that the dimension $d = 79$ is necessary to capture all the information. We hypothesize that this might be due to the fact that the expression parameters $\boldsymbol{e}$ are extracted from a 3DMM, after applying PCA on large human face datasets and keeping the most important principal components. This would be an interesting avenue to explore for future work.

---

[3] https://numpy.org/doc/stable/reference/generated/numpy.linalg.svd.html

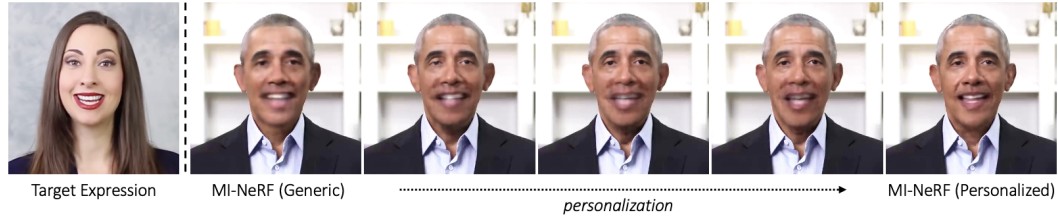

Target Expression    MI-NeRF (Generic)    ······················*personalization*······················→    MI-NeRF (Personalized)

Figure 7: Progressive **personalization** from our generic MI-NeRF trained on 100 identities to the personalized model for a target identity.

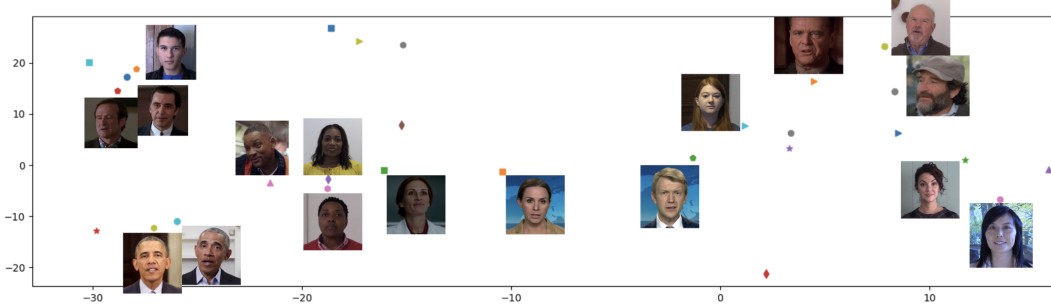

Figure 8: A visualization of our learned identity codes using t-SNE (Van der Maaten & Hinton, 2008). We notice that the identity codes capture meaningful information in terms of identity, e.g. different videos of Obama (bottom left) are clustered together, despite the different lighting.

## C    ADDITIONAL RESULTS

In this section, we include additional qualitative and quantitative results, in order to further evaluate our method.

**Personalization.** Fig. 7 shows the progressive improvement of the visual quality during our personalization procedure (see Sec. 3.4). Compared to the generic MI-NeRF, the personalized model better captures the high-frequency facial details, such as wrinkles.

**Learned Identity Codes.** Fig. 8 shows a visualization of our learned identity codes using t-SNE (Van der Maaten & Hinton, 2008) on our final model with 100 identities. We notice that the identity codes capture meaningful information in terms of identity.

**Learned Latent Codes.** Fig. 10 shows a qualitative comparison between our model trained without and with latent codes $l$ for the same expression as input. As mentioned in the ablation study in Sec. 4.2, these latent codes capture very small variations in appearance and high-frequency details.

**Additional Qualitative Results.** Fig. 9 demonstrates additional qualitative results of facial expression transfer generated by MI-NeRF. Fig. 11 shows additional comparison with GeneFace Ye et al. (2023) for lip synced video synthesis.

**GAN-based Lip Synced Video Synthesis.** As also described in Chatziagapi et al. (2023), Wav2Lip (Prajwal et al., 2020) is a state-of-the-art GAN-based model for talking face video synthesis, trained on thousands of videos. It can generate well-synchronized lip movements for any target audio. However, it can produce blurry results and artifacts in the mouth region, particularly visible in HD resolution. These artifacts frequently appear in large head movements and expressive faces (see Fig. 4 and suppl. video). Thus, it leads to low visual quality (PSNR 28.90, SSIM, 0.89, LPIPS 0.17 - see Table 2 right and Chatziagapi et al. (2023)). As a GAN network that operates in the 2D image space, it does not model the 3D face geometry. On the other hand, NeRFs have demonstrated high-quality 3D modeling of the human face. But, since NeRFs are trained on a single-identity video of limited duration, they cannot learn as good lip synchronization - Wav2Lip achieves LSE-D/LSE-C 8.06/4.77 (see Table 2 and Chatziagapi et al. (2023)). Please note the Wav2Lip is optimized for

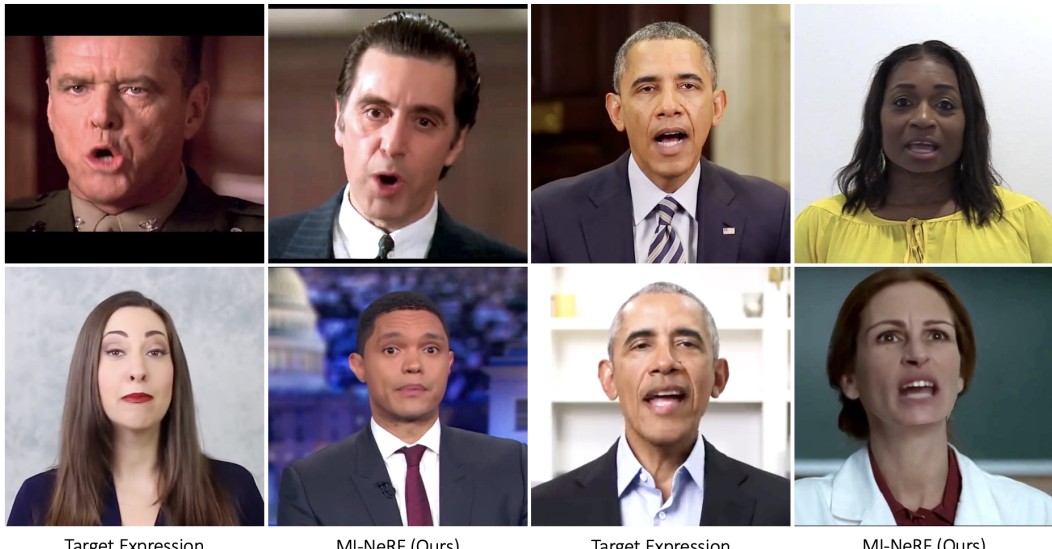

Figure 9: Facial expression transfer for various identities by our proposed MI-NeRF.

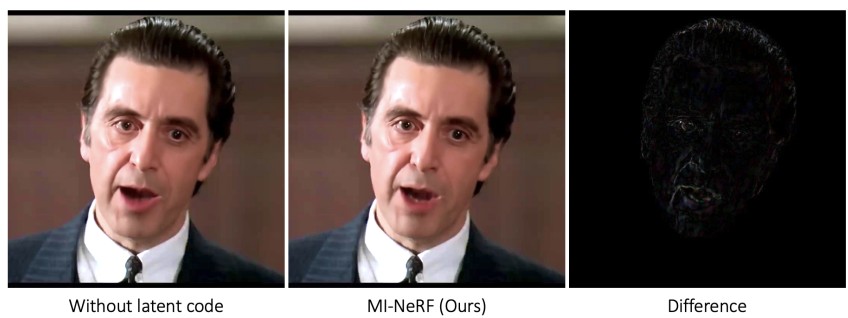

Figure 10: Ablation study on our learned latent codes. From left to right: result of MI-NeRF with $M$ without latent codes $l$, result of MI-NeRF (Ours), and their difference.

these metrics during training. With our work, we go one step forward towards training a NeRF on thousands of videos, similarly to GANs. In this way, we can leverage the high-quality 3D face modeling of NeRFs and information from multiple identities.

**Video Results.** We strongly encourage the readers to watch our supplementary video that include results for facial expression transfer and lip synced video synthesis.

## D    IMPLEMENTATION DETAILS

In this section, we include additional implementation details. We closely follow the architecture and training details of NeRFace (Gafni et al., 2020), AD-NeRF (Guo et al., 2021), and LipNeRF (Chatziagapi et al., 2023) that are all similar. An overview of the main hyper-parameters is given in Table 4. More specifically, our implementation is based on PyTorch (Paszke et al., 2019). We use Adam optimizer (Kingma & Ba, 2014) with a learning rate that begins at $5 \times 10^{-4}$ and decays exponentially to $5 \times 10^{-5}$ during training. The rest of the Adam hyper-parameters are set at their default values ($\beta_1 = 0.9, \beta_2 = 0.999, \epsilon = 10^{-8}$). For every gradient step, we march rays and sample points for a single randomly-chosen video frame. We march 2048 rays and sample 64 points per ray for the coarse volume and 128 points per ray for the fine volume (hierarchical sampling strategy (Mildenhall et al., 2020)). We sample rays such that $95\%$ of them correspond to pixels inside the detected bounding box of the head (Gafni et al., 2020). We also assume that the last sample on each ray lies on

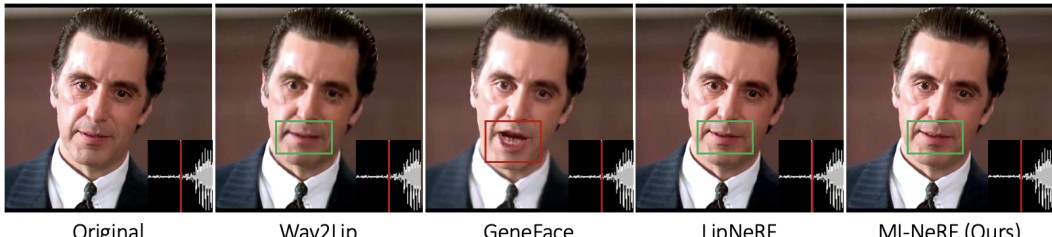

| Original | Wav2Lip | GeneFace | LipNeRF | MI-NeRF (Ours) |

Figure 11: Qualitative comparison of lip synced video synthesis by Wav2Lip (Prajwal et al., 2020), GeneFace (Ye et al., 2023), LipNeRF (Chatziagapi et al., 2023), and MI-NeRF. The original video is in English (1st column). The generated videos (columns 2-5) are lip synced to dubbed audio in Spanish.

| | |
|---|---|
| Optimizer | Adam |
| Initial learning rate | $5 \times 10^{-4}$ |
| Final learning rate | $5 \times 10^{-5}$ |
| Learning rate schedule | exponential decay |
| Batch size (number of rays) | 2048 |
| Samples for the coarse network (per ray) | 64 |
| Samples for the fine network (per ray) | 128 |
| Linear layers (MLP backbone) | 8 |
| Hidden units (MLP backbone) | 256 |
| Activation | ReLU |
| Frequencies for $x$ (positional encoding) | 10 |
| Frequencies for $v$ (positional encoding) | 4 |

Table 4: **Experimental setting**. Hyper-parameters used for training our proposed MI-NeRF (used for both facial expression transfer and talking face video synthesis).

the background and takes the corresponding RGB color (Guo et al., 2021; Chatziagapi et al., 2023). We use the original background per frame. Our MLP backbone consists of 8 linear layers with 256 hidden units each. The output of the backbone is fed to an additional linear layer to predict the density $\sigma$, and a 4-layer 128-unit wide branch to predict the RGB color $c$ for every point. We use ReLU activations. Positional encodings are applied to both the input points $x$ and the viewing directions $v$, of 10 and 4 frequencies respectively. We do not apply smoothing on the expression parameters using a low pass filter, as proposed by Chatziagapi et al. (2023). Instead, we learn a self-attention, similarly to Guo et al. (2021), that is applied to both the output of the multiplicative module and the latent codes after the first 200k iterations during training. This ensures smooth results in the final video synthesis, reducing any jitter. For 1-2 identities, we train our network for about 400k iterations (around 40 hours on a single GPU). For 10 or 20 identities, we need around 500k and 600k iterations correspondingly, and our generic MI-NeRF with 100 identities takes about 800k iterations (see Fig. 3). Further personalization requires another 50-80k iterations approximately for a target identity.

To compute the visual quality metrics (PSNR, SSIM, LPIPS), we crop each frame around the face, using the face detector from 3DDFA (Guo et al., 2020; 2018a). In this way, we evaluate the visual quality of the generated part only, ignoring the background that corresponds to the original one. To verify the speaker's identity, we use the ACD metric, computing the cosine distance between the embeddings of the ground truth face and the generated face, extracted by InsightFace [4].

## E  ETHICAL CONSIDERATIONS

We would like to note the potential misuse of video synthesis methods. With the advances in neural rendering and generative models, it becomes easier to generate photorealistic fake videos of any identity. These can be used for malicious purposes, e.g. to generate misleading content and spread

---
[4]https://github.com/deepinsight/insightface

misinformation. Thus, it is important to develop accurate methods for fake content detection and forensics. Research on discriminative tasks has been investigated for several years by the community and there are certain guarantees and knowledge on how to build strong classifiers. However, this is only the first step towards mitigating the issue of fake content; further steps are required. A possible solution, that can be easily integrated in our work, is watermarking the generated videos (Chen et al., 2023), in order to indicate their origin. In addition, appropriate procedures must be followed to ensure fair and safe use of videos from a social and legal perspective.

# F   DATASET DETAILS

In this section, we provide additional details for the dataset we used in our experiments. As mentioned in Sec. 4.1, we collected 140 talking face videos from publicly available datasets, which are commonly used in related works (Guo et al., 2021; Lu et al., 2021; Chatziagapi et al., 2023; Hazirbas et al., 2021; Ginosar et al., 2019; Ahuja et al., 2020; Duarte et al., 2021; Zhang et al., 2021; Wang et al., 2021b). The detailed list of the 140 videos is as follows:

- Standard videos used in related works, collected by Lu et al. (2021) and Guo et al. (2021):
    - Obama1 (Barack Obama)
    - Obama2 (Barack Obama)
    - Markus Preiss
    - Natalie Amiri
- PATS dataset (Ginosar et al., 2019; Ahuja et al., 2020):
    - Trevor Noah
    - John Oliver
    - Samantha Bee
    - Charlie Houpert
    - Vanessa Van Edwards
- Actors from dataset proposed by LipNeRF (Chatziagapi et al., 2023):
    - Al Pacino
    - Jack Nicholson
    - Julia Roberts
    - Robin Williams
    - Tom Hanks
    - Morgan Freeman
    - Tim Robbins
    - Will Smith
- TalkingHead-1KH dataset (Wang et al., 2021b):
    - 1lSejjfNHpw_0075_S0_E1456_L671_T47_R1471_B847
    - 2Xu56MEC91w_0046_S80_E1105_L586_T86_R1314_B814
    - 3y6Vjr45I34_0004_S287_E1254_L568_T0_R1464_B896
    - 4hQi42Q9mcY_0002_S0_E1209_L443_T0_R1515_B992
    - 5crEV5DbRyc_0009_S208_E1152_L1058_T102_R1712_B756
    - -7TMJtnhiPM_0000_S1202_E1607_L345_T26_R857_B538
    - -7TMJtnhiPM_0000_S1608_E1674_L467_T52_R851_B436
    - 85UEFVcmIjI_0014_S92_E1162_L558_T134_R1294_B870
    - A2800grpOzU_0002_S812_E1407_L227_T7_R1139_B919
    - c1DRo3tPDG4_0010_S0_E1730_L432_T33_R1264_B865
    - EGGsK7po68c_0007_S0_E1024_L786_T50_R1598_B862
    - eKFlMKp9Gs0_0005_S0_E1024_L705_T118_R1249_B662
    - EWKJprUrnPE_0005_S0_E1024_L84_T168_R702_B786
    - gp4fg9PWuhM_0003_S0_E858_L526_T0_R1310_B768

- – HBlkinewdHM_0000_S319_E1344_L807_T149_R1347_B689
- – jpCrKYWjYD8_0002_S0_E1535_L527_T68_R1215_B756
- – jxi_Cjc8T1w_0061_S0_E1024_L660_T102_R1286_B728
- – kMXhWN71Ar0_0001_S0_E1311_L60_T0_R940_B832
- – m2ZmZflLryo_0009_S0_E1024_L678_T51_R1390_B763
- – NXpWIephX1o_0031_S0_E1264_L357_T0_R1493_B1072
- – PAaWZTFRP9Q_0001_S0_E672_L624_T42_R1376_B794
- – PAaWZTFRP9Q_0001_S926_E1425_L696_T101_R1464_B869
- – SmtJ5Cy4jCM_0006_S0_E523_L524_T50_R1388_B914
- – SmtJ5Cy4jCM_0006_S546_E1134_L477_T42_R1357_B922
- – SU8NSkuBkb0_0015_S826_E1397_L347_T69_R1099_B821
- – VkKnOEQlwl4_0010_S98_E1537_L821_T22_R1733_B934
- – –Y9imYnfBw_0000_S0_E271_L504_T63_R792_B351
- – –Y9imYnfBw_0000_S1015_E1107_L488_T23_R824_B359
- – YsrzvkG5_KI_0018_S36_E1061_L591_T100_R1055_B564
- – Zel-zag38mQ_0001_S0_E1466_L591_T12_R1439_B860
- Casual Conversations dataset (Hazirbas et al., 2021):
  - – 1224_09
  - – 1226_00
  - – 1229_08
  - – 1230_09
  - – 1232_00
  - – 1233_09
  - – 1234_11
  - – 1235_09
  - – 1247_00
  - – 1249_14
  - – 1250_09
  - – 1253_00
  - – 1269_11
  - – 1281_06
  - – 1281_13
  - – 1282_10
  - – 1290_07
  - – 1290_13
  - – 1301_11
  - – 1323_09
  - – 1328_14
- How2Sign dataset (Duarte et al., 2021):
  - – 0zvsqf23tmw_3-2-rgb_front
  - – 2ri5HYm48MA_5-2-rgb_front
  - – 4I2azcR2kcA-8-rgb_front
  - – 5Uy3r6Sl4pM-8-rgb_front
  - – 5z_z6opEIH0-3-rgb_front
  - – -96cWDhR4hc-5-rgb_front
  - – a1HVL0zE768_2-3-rgb_front
  - – a4Nxq0QV_WA_5-5-rgb_front
  - – bIUmw2DVW7Q_11-3-rgb_front
  - – dlXnxaYWr9w-1-rgb_front
- HDTF dataset (Zhang et al., 2021):

- RD_Radio10_000
- RD_Radio1_000
- RD_Radio11_000
- RD_Radio11_001
- RD_Radio12_000
- RD_Radio13_000
- RD_Radio14_000
- RD_Radio16_000
- RD_Radio17_000
- RD_Radio18_000
- RD_Radio19_000
- RD_Radio20_000
- RD_Radio2_000
- RD_Radio21_000
- RD_Radio22_000
- RD_Radio23_000
- RD_Radio25_000
- RD_Radio26_000
- RD_Radio27_000
- RD_Radio28_000
- RD_Radio29_000
- RD_Radio30_000
- RD_Radio3_000
- RD_Radio31_000
- RD_Radio32_000
- RD_Radio33_000
- RD_Radio34_000
- RD_Radio34_001
- RD_Radio34_002
- RD_Radio34_003
- RD_Radio34_004
- RD_Radio34_005
- RD_Radio34_006
- RD_Radio34_007
- RD_Radio34_009
- RD_Radio35_000
- RD_Radio36_000
- RD_Radio37_000
- RD_Radio38_000
- RD_Radio39_000
- RD_Radio40_000
- RD_Radio4_000
- RD_Radio41_000
- RD_Radio42_000
- RD_Radio43_000
- RD_Radio44_000
- RD_Radio45_000
- RD_Radio46_000
- RD_Radio47_000
- RD_Radio48_000

- – RD_Radio49_000
- – RD_Radio50_000
- – RD_Radio5_000
- – RD_Radio51_000
- – RD_Radio52_000
- – RD_Radio53_000
- – RD_Radio54_000
- – RD_Radio57_000
- – RD_Radio59_000
- – RD_Radio7_000
- – RD_Radio8_000
- – RD_Radio9_000

Please note that most of these videos are from YouTube and the identities are public figures (e.g. politicians in HDTF dataset (Zhang et al., 2021), famous actors in LipNeRF Chatziagapi et al. (2023), comedians and professional YouTubers in PATS dataset (Ginosar et al., 2019; Ahuja et al., 2020) [5]). The Casual Conversations dataset (Hazirbas et al., 2021) includes talking videos of paid individuals who agreed to participate and opted-in for data use in ML. The participants are de-identified with unique numbers [6]. The How2Sign dataset (Duarte et al., 2021) is another dataset publicly available for research purposes that includes American Sign Language videos [7]. TalkingHead-1kH (Wang et al., 2021b) also includes videos from YouTube under permissive licenses only [8].

In Sec. 4.1, we mention that we use 100 videos for training our multi-identity model. These correspond to 100 different identities, where we use both Obama1 and Obama2 (see list above) that are different videos of Obama (see also Fig. 8). Thus, more specifically, we use 101 videos in total for training and we learn 101 identity codes. We mention "100 identities" for simplicity purposes.

For each video, we fit a 3DMM, as described in Sec. 3.1, in order to extract the corresponding pose and expression parameters of the identity per frame. We refer the interested reader to the work of Guo et al. (2021) [9] for more details in the data preprocessing.

---

[5] https://chahuja.com/pats/
[6] https://ai.meta.com/datasets/casual-conversations-dataset/
[7] https://how2sign.github.io/
[8] https://github.com/tcwang0509/TalkingHead-1KH
[9] https://github.com/YudongGuo/AD-NeRF

