# OpenReview forum: "MI-NeRF: Learning a Single Face NeRF from Multiple Identities"
_ICLR.cc/2024/Conference — Submitted to ICLR 2024_

### Official Review · Reviewer_x1A2 · 2023-10-31

**Soundness:** 3 good
**Presentation:** 3 good
**Contribution:** 3 good
**Rating:** 8
**Confidence:** 4

**Summary:**

The paper introduces MI-NeRF, a novel approach to learn a single dynamic neural radiance field (NeRF) from monocular videos of talking faces across multiple identities. The core innovation is the incorporation of a multiplicative module that distinguishes between identity and expression information, inspired by TensorFaces. By simultaneously training on multiple videos, MI-NeRF significantly cuts training time while achieving state-of-the-art performance in tasks like facial expression transfer and talking face video synthesis.

**Strengths:**

1. The proposed MI-NeRF robustly adapts to unseen expressions and identities, requiring minimal retraining, thereby saving computational time and effort.
2. The Multiplicative module in MI-NeRF effectively separates identity from expression, ensuring consistent identity portrayal even with dynamic expressions, a challenge for some other models.
3. MI-NeRF achieves superior performance in visual quality and lip synchronization, making it a leading solution in realistic talking face synthesis.

**Weaknesses:**

Here are some concerns:

1. The paper utilizes learnable latent codes to capture time-varying information. However, there's a potential ambiguity regarding how the model ensures these codes don't inadvertently encode identity or expression information. A more rigorous analysis or mechanism would have been beneficial to validate that these latent codes truly only represent unique, time-varying elements without overlapping with identity and expression descriptors.

2. Maintaining consistency in novel views is inherently tied to the accurate modeling of 3D information. When MI-NeRF constructs facial identities and expressions over time, it is imperative for it to ensure that these reconstructions remain consistent and accurate, even when viewed from unseen angles. This is not merely a question of aesthetics, but of the system's fidelity to real-world dynamics. Without evaluation or visualization of its ability to maintain this consistency, one could question its versatility and applicability in varied real-world settings.

3. Some concerns regarding the dataset:
i) The collected videos lead to concerns about data privacy and the feasibility of public release. Ethical considerations surrounding personal data must be paramount. And will the authors be able or allowed to release the data?
ii) Within 140 videos, It's unclear how many unique individuals are represented and whether multiple videos pertain to the same person. Additionally, questions arise regarding the consistency of identity codes across different videos during training.
iii) The steps taken to prepare the data before its integration into MI-NeRF are essential. Without clear details about preprocessing, it's challenging to replicate results or trust the dataset's integrity.

4. While the research highlights the limitations of the GAN-based method, Wav2Lip, it doesn't delve deeply into how MI-NeRF stands compared to other recent state-of-the-art GAN approaches.

5. MI-NeRF's uniqueness largely stems from its multiplicative module. However, this also means the model's overall success heavily depends on this single component. If the module faces challenges in complex real-world scenarios, the entire model's performance could be compromised. A deeper discussion about the potential limitations of this module would be beneficial for a comprehensive understanding of its robustness.

**Questions:**

1. Could the authors shed light on the exact nature of the per-frame latent codes and how they ensure these codes are devoid of identity or expression specifics?

2. Could there be a more detailed comparison between MI-NeRF and advanced GAN-based methods, particularly in the context of face video synthesis?

---

> ### Author Response · Authors · 2023-11-18
> **Response to reviewer x1A2**
>
> We are thankful to the reviewer x1A2 for their effort and time to review our paper. We address their concerns below:
>
>
> > Q1: Learned latent codes.
>
> We included in the appendix C a qualitative comparison of our model trained with and without the latent codes (see [Fig. 10](https://imgur.com/a/FkYTlf0)).  In the revised manuscript, we denote with red color the changes for visual distinction. As the indicative frame illustrates, our learned latent codes only capture very small variations in appearance and high-frequency details, while the expression and the identity remain the same (see also ablation study in Sec. 4.2). Thus, our experimental analysis exhibits that there is no overlap of these codes with identity and expression.
>
> In practice, we ensure this by adding a higher L2 regularization on the latent codes (see $\lambda_{l}$ in Sec. 3.3). This results in latent codes only obtaining small values and capturing very small per-frame variations, that are not captured by the powerful representation of the expression parameters in the PCA space, or the time-invariant identity code.
>
>
> > Q2: Comparison with GAN-based methods.
>
> Earlier works that propose GAN-based methods for expression transfer, like GANimation [1], operate in the 2D image space. Similarly with Wav2Lip, these 2D-based approaches frequently produce artifacts, since they cannot handle 3D motion (see Fig. 4, Sec. 4.4, and appendix C).
>
> Recent state-of-the-art GANs propose 3D-aware approaches, like Next3D [2], that are based on EG3D [3] and use StyleGAN2 [4] priors. The output identity depends on the input latent code and therefore on the random seed. In order to render a specific person, they require to run a time-consuming GAN inversion (e.g. PTI [5]). This process is required in order to project the true identity to the closest identity the model can express. To achieve that, they use the closest latent codes and fine-tune the generator weights, in order to approximate an identity, given their image. In contrast, in our case, we can directly learn a code for each identity and render them with a high fidelity. In addition, these 3D-aware GANs *require very long training times* (e.g. training EG3D for 8.5 days on 8 Tesla V100 GPUs [3] and another 4 days training for Next3D on 4 3090 GPUs [2]).
>
>
> > Q3: Dataset details.
>
> We included in the pdf an additional section in the appendix, Sec. F, where we provide additional details for the dataset. As Sec. F clarifies, we collected talking face videos from publicly available datasets, commonly used by other published works. We provide a detailed list of the identities and videos used, as well as the corresponding links that include the origin of the datasets and the license. A visualization of the learned identity codes is given in Fig. 8.
>
> As mentioned in Sec. 3.1 and Sec. F, we run standard pre-processing techniques to fit a 3DMM per video frame and extract the corresponding head pose and expression parameters. We refer the reviewer to [6] for more details in the data pre-processing.
>
> If the reviewer has any remaining concerns, we would be happy to address them.
>
>
> > Q4. Limitations of the multiplicative module.
>
> Our proposed multiplicative module learns non-linear interactions between identity and expression. To learn these interactions, we use talking face videos that include a variety of facial expressions from different identities. We believe that a limitation would arise if there is no overlap between the expressions of different identities. As shown in [7], multiplicative interactions can be successfully captured if there is some overlap between different attributes, in order to learn all possible combinations.  However, the case of no overlap would not easily appear in a real-world scenario, since it is very probable that different subjects will show some similar expressions, e.g. neutral expression or a smile, while talking.
>
>
> ___
> ## References
>
> [1] Pumarola et al., GANimation: One-Shot Anatomically Consistent Facial Animation, IJCV, 2019.
>
> [2] Sun et al., Next3D: Generative Neural Texture Rasterization for 3D-Aware Head Avatars, CVPR, 2023.
>
> [3] Chan et al., Efficient Geometry-aware 3D Generative Adversarial Networks, CVPR, 2022.
>
> [4] Karras et al., Analyzing and improving the image quality of StyleGAN, CVPR, 2020.
>
> [5] Roich et al., Pivotal tuning for latent-based editing of real images, ACM Trans. Graph., 2021.
>
> [6] Guo et al., Cnn-based real-time dense face reconstruction with inverse-rendered photo-realistic face images, IEEE transactions on pattern analysis and machine intelligence, 2018.
>
> [7] Georgopoulos et al., Multilinear latent conditioning for generating unseen attribute combinations, ICML, 2020.

---

### Official Review · Reviewer_ud4c · 2023-11-01

**Soundness:** 3 good
**Presentation:** 4 excellent
**Contribution:** 3 good
**Rating:** 6
**Confidence:** 3

**Summary:**

In this paper, authors propose a NeRF based method for dynamic face synthesis from monocular talking face videos. Unlike single-identity models, the proposed method uses a single network to train on videos of multiple identities and do a fast fine tuning when applying on a new identity. This single network is trained by imposing a multiplicative structure between identity embeddings and expression embeddings.  Quantitative and qualitative experiments show that the proposed multiplicative structure helps disentangling the identity and expression. The training time is significantly reduced compared to single-identity models.

**Strengths:**

The proposed method is simple and effective.

Built on top of earlier multi-identity NeRF methods, this work basically imposes a multiplicative structure between identity embeddings and expression embeddings. The structure is in the form of an element-wise product between two embeddings and can be extended with high-degree interactions. The method shouldn't be hard for any readers to re-implement.

Ablation studies show the multiplicative structure is well designed, and each piece of it can help improve the factor disentanglement and visual quality. When compared with existing works, the proposed method shows superior performance for expression transfer and lip synced video synthesis tasks.

**Weaknesses:**

The comparison with other multi-identity NeRF-based methods can be improved.

Fast training time is a known benefit for multi-identity NeRF-based methods. For the training time evaluation, it misses the comparison with other multi-identity methods, like HeadNeRF.

More multi-identity NeRF-based methods might be included in the quantitative comparison, especially in the expression transfer experiments.

**Questions:**

Is there a training time benchmarking (Figure 3) for identities methods, like HeadNeRF?

---

> ### Author Response · Authors · 2023-11-13
> **Response to reviewer ud4c**
>
> We are thankful to the reviewer ud4c for their time and effort to review our paper.
>
> > Q1: Comparison with other multi-identity NeRF-based methods.
>
> We believe that we have included all the relevant qualitative and quantitative comparisons with other multi-identity NeRF methods. As mentioned in our paper, to the best of our knowledge, MI-NeRF is the first multi-identity NeRF for faces, learned from monocular videos of multiple subjects. For facial expression transfer, we show comparisons with the single-identity NeRFace, its immediate extension "Baseline NeRF" trained on multiple identities, and the NeRF-based parametric model HeadNeRF (see Sec. 4.3, Fig. 2, Table 2). However, if the reviewer ud4c is aware of other multi-identity NeRF-based methods and would like to see a comparison with those, please let us know.
>
> __________
>
> > Q2: Fast training time is a known benefit for multi-identity NeRF-based methods. For the training time evaluation, it misses the comparison with other multi-identity methods, like HeadNeRF.
>
> According to the respective paper, HeadNeRF requires about 3 days of training on a single NVIDIA 3090 GPU, which is similar to our reported time for training of 100 identities.
>
> If the reviewer has any remaining questions, we are happy to address them. We would appreciate it if the reviewer provides any concrete multi-identity NeRFs they believe we should compare with.

---

### Official Review · Reviewer_TTon · 2023-11-05

**Soundness:** 3 good
**Presentation:** 3 good
**Contribution:** 3 good
**Rating:** 6
**Confidence:** 3

**Summary:**

This paper proposes a method to learn a single dynamic NeRF for talking face videos of multiple identities. Expression, identity, and time-varying parts are separately modeled and interact with each other to predict the color and density of NeRF. Therefore, this method can use only a single NeRF to model the common geometry for diverse faces.

**Strengths:**

1. The whole writing is well organized and the proposed method is clearly described;
2. The method for disentangling the identity and expression sounds reasonable, and the experiments show good disentanglement;
3. Shown results outperform the competitors and the training time is significantly reduced.

**Weaknesses:**

The paper further proposes a high-degree interaction module; however, I don't see any usage of this module in the whole method, as well as any experiment analysis about this module.

**Questions:**

Please refer to the weakness.

---

> ### Author Response · Authors · 2023-11-15
> **Response to reviewer TTon**
>
> We are thankful to the reviewer TTon for their time and effort to review our paper. We address their concern below:
>
>
> > Q1: High-degree module.
>
> We conduct experiments using the high-degree module H, please see Table 1. The results indicate that it performs on par with the module M that we propose and visually we also observe a similar visual quality to the module M.
>
> In addition, this module can be even more useful when interactions between additional variables should be captured, e.g., if we include lighting conditions in the future. Therefore, we do believe that this method is useful in our methodological part.

---

### Official Review · Reviewer_d6cv · 2023-11-08

**Soundness:** 3 good
**Presentation:** 2 fair
**Contribution:** 2 fair
**Rating:** 6
**Confidence:** 4

**Summary:**

This work is about developing a single NeRF for multiple face identities. It uses the 3D morphable face model to estimte the 3D first, and then applys to NeRF for building the model. Experiments are shown both quantitatively and quanlitatively.

**Strengths:**

It argues that a single face NeRF model can be developed for multiple identities.

Some interesting results are obtained.

**Weaknesses:**

It is unclear how the face expressions are aligned or handled. Even different identities can be handled together, how to deal with the different expressions for different identities? Do you use a cononical face model to separate the expressions?

It is unclear how to handle diferent lighting conditions.

There is no quantitative comparisons between the proposed method and the state of the art. It is unclear if the proposed method can outperform the existing works.

**Questions:**

As listed in the weakness part.

Further, the authors should give clearer statements on how to deal with different identities for the NeRF model learning. Although some equations are given, it is still difficult how to do this. It needs more detailed descriptions on this point.

**Details Of Ethics Concerns:**

For face images, there are some privacy issues to address.

---

> ### Author Response · Authors · 2023-11-15
> **Response to reviewer d6cv**
>
> We are thankful to the reviewer d6cv for their time and effort to review our paper. We address their concerns below:
>
>
> > Q1:  How the face expressions are aligned or handled.
>
> We perform standard processing techniques. Please let us elaborate on those. As described in Sec. 3.1, we fit a 3D morphable model (3DMM) for each video frame, and extract the corresponding head pose and facial expression parameters. A 3DMM [1] is a parametric model that represents  the human face as a linear combination of principle axes for shape, texture, and expression, learned by principal components analysis (PCA). Thus, the extracted expression coefficients are in a common space for different identities. The extracted head pose corresponds to rotation and translation matrices that are used to transform the sampled 3D points to the canonical space (see Sec. 3.1). We also refer the reviewer to the cited work of [2]  for more details.
>
>
> > Q2: Lighting conditions.
>
> Handling different lighting conditions is an interesting research problem, but beyond the scope of this paper. We refer the reviewer to related works that address this problem [3, 4]. In our case, in addition to expression and identity codes, we learn per-frame latent codes (embeddings) that capture time-varying information (see Sec. 3.1). These codes are used to capture small per-frame variations in appearance and illumination, and reconstruct them in the generated videos, in order to enhance the final visual quality (see ablation study in Sec. 4.2). Since our training videos do not include any significant lighting changes, these latent codes only give a very small increase in performance. Handling larger lighting changes is an interesting problem for future research.
>
>
> > Q3: Quantitative comparisons.
>
> We provide extensive quantitative comparisons with the relevant state-of-the-art methods in Sec. 4.3 and 4.4, and in Table 2. If the reviewer d6cv is aware of any other relevant state-of-the-art work and would like to see a comparison with those, please let us know.
>
> > Q4: How to deal with different identities.
>
> As described in Sec. 3.1, we learn an identity code per video. This is a learnable embedding that is optimized during training (see torch.nn.Embedding from PyTorch). By enforcing it to be the same for all the frames of a subject, we encourage it to capture time-invariant information. These identity codes are used in the proposed multiplicative module (see Sec. 3.2).
>
>
> > Q5: With respect to ethics review.
>
> We included in the pdf an additional section in the appendix, Sec. F, where we provide additional details for the dataset. The changes in the manuscript are highlighted with red for visual clarity. As Sec. F clarifies, we collected talking face videos from publicly available datasets, commonly used by other published works [5, 6, 7, 8, 9]. We provide a detailed list of the identities and videos used, as well as the corresponding links that include the origin of the datasets and the license.
>
> In addition, our original submission does include an ethics statement as encouraged by the ICLR guidelines. If the reviewer believes something else should be added there, we would be happy to include it.
>
> If the reviewer has any remaining concerns, we would be happy to address them.
>
> ____
>
> ## References
>
> [1] Blanz et al., A morphable model for the synthesis of 3D faces, Proceedings of the 26th annual conference on computer graphics and interactive techniques, 1999.
>
> [2] Guo et al., Cnn-based real-time dense face reconstruction with inverse-rendered photo-realistic face images, IEEE transactions on pattern analysis and machine intelligence, 2018.
>
> [3] Wang et al., Portrait Reconstruction and Relighting using the Sun as a Light Stage, CVPR, 2023.
>
> [4] Jiang et al., NeRFFaceLighting: Implicit and Disentangled Face Lighting Representation Leveraging Generative Prior in Neural Radiance Fields, ACM Transactions on Graphics, 2023.
>
> [5] Guo et al., AD-NeRF: Audio driven neural radiance fields for talking head synthesis, CVPR, 2021.
>
> [6] Lu et al., Live Speech Portraits: Real-Time Photorealistic Talking-Head Animation, ACM Transactions on Graphics, 2021.
>
> [7] Chatziagapi et al., LipNeRF: What is the right feature space to lip-sync a NeRF, International Conference on Automatic Face and Gesture Recognition, 2023.
>
> [8] Zhang et al., Flow-Guided One-Shot Talking Face Generation With a High-Resolution Audio-Visual Dataset, CVPR, 2021.
>
> [9] Liu et al., Semantic-Aware Implicit Neural Audio-Driven Video Portrait Generation, ECCV, 2022.

---

### Author Response · Authors · 2023-11-21
**Please let us know if you have any remaining questions**

Dear reviewers and AC(s),

We are thankful to all of you for your time and effort to handle our paper and eventually help us improve it. We are glad that the reviewers find our method interesting and technically sound, and for acknowledging the good disentanglement between identity and expression with our proposed multiplicative module, the improvement over the current state-of-the-art, as well as the significant decrease in the total training time.

We have replied to their individual comments and concerns. Since the discussion window is closing soon, please let us know if there are any remaining questions or concerns from the perspective of the reviewers.

Once again, we are grateful for your feedback so far.

Best,

Authors

---

### Meta-Review · Area_Chair_KZhG · 2023-12-05

**Metareview:**

This paper investigates how to learn a unified neural radiance field (NeRF) from videos of different people. It presents a Multi-identity NeRF(MI-NeRF) network with modules for learning disentangled identity and non-identity features.
 The experiments on facial expression transfer and talking face synthesis demonstrate the effectiveness of the proposed method.

Strengths:
+ The idea of multi-identity NeRF is interesting.
+ The proposed network for disentangling identity and expression is reasonable.
+ The proposed method significantly reduces the training time.
+ The experiments show the improvement over the current state-of-the-art.
+ The paper is well-written and easy to follow.

Weaknesses:
- There are some limitations with the proposed  multiplicative module-- it requires overlap between the expressions of different identities.
- Some details are missing: It is unclear how the face expressions are aligned or handled. It is unclear how to handle different lighting conditions.
- Miss comparisons between the proposed method and some related work.
- The paper does not show the usage of the proposed high-degree interaction module.

**Justification For Why Not Higher Score:**

This is a borderline paper. Reviewers acknowledge that this investigation is worthwhile. However, there are shared concerns on the limitations with the proposed method, missing details and experiments.
The authors address some of these, providing additional experiments and arguments, but these were not enough to sway reviewers. I think this paper is not ready for publication at the current stage, and give the authors more time to improve the paper.

**Justification For Why Not Lower Score:**

N/A

---

### Decision · Program_Chairs · 2024-01-16

Reject